# The Association Among Bipolar Disorder, Mitochondrial Dysfunction, and Reactive Oxygen Species

**DOI:** 10.3390/biom15030383

**Published:** 2025-03-06

**Authors:** Yuki Kageyama, Shohei Okura, Ayaka Sukigara, Ayaka Matsunaga, Kunio Maekubo, Takafumi Oue, Koki Ishihara, Yasuhiko Deguchi, Koki Inoue

**Affiliations:** Department of Neuropsychiatry, Osaka Metropolitan University Graduate School of Medicine, Osaka 545-8585, Japanv22028k@omu.ac.jp (A.S.); w21235k@omu.ac.jp (Y.D.);

**Keywords:** bipolar disorder, inflammation, mitophagy, synaptic plasticity, reactive oxygen species

## Abstract

Mitochondria, often known as the cell’s powerhouses, are primarily responsible for generating energy through aerobic oxidative phosphorylation. However, their functions extend far beyond just energy production. Mitochondria play crucial roles in maintaining calcium balance, regulating apoptosis (programmed cell death), supporting cellular signaling, influencing cell metabolism, and synthesizing reactive oxygen species (ROS). Recent research has highlighted a strong link between bipolar disorder (BD) and mitochondrial dysfunction. Mitochondrial dysfunction contributes to oxidative stress, particularly through the generation of ROS, which are implicated in the pathophysiology of BD. Oxidative stress arises when there is an imbalance between the production of ROS and the cell’s ability to neutralize them. In neurons, excessive ROS can damage various cellular components, including proteins in neuronal membranes and intracellular enzymes. Such damage may interfere with neurotransmitter reuptake and the function of critical enzymes, potentially affecting brain regions involved in mood regulation and emotional control, which are key aspects of BD. In this review, we will explore how various types of mitochondrial dysfunction contribute to the production of ROS. These include disruptions in energy metabolism, impaired ROS management, and defects in mitochondrial quality control mechanisms such as mitophagy (the process by which damaged mitochondria are selectively degraded). We will also examine how abnormalities in calcium signaling, which is crucial for synaptic plasticity, can lead to mitochondrial dysfunction. Additionally, we will discuss the specific mitochondrial dysfunctions observed in BD, highlighting how these defects may contribute to the disorder’s pathophysiology. Finally, we will identify potential therapeutic targets to improve mitochondrial function, which could pave the way for new treatments to manage or mitigate symptoms of BD.

## 1. Background

Mitochondria are organelles in nearly all eukaryotic cells and are the primary site for cellular energy production. Their main function is to convert nutrients into adenosine triphosphate (ATP), the energy required by the cell. In addition, mitochondria have multiple functions, such as reactive oxygen species (ROS) metabolism, calcium (Ca^2+^) homeostasis, and cell death and survival. The ATP synthesis system in the mitochondrial inner membrane comprises five enzyme complexes (respiratory chain complexes I to V). NADH and FADH_2_, produced via glycolysis and the citric acid cycle, are oxidized by respiratory chain complexes I to IV in the mitochondrial inner membrane. This oxidation involves electron transfer and active proton transport, ultimately reducing oxygen to water. Complex V synthesizes ATP using this proton motive force, a process known as oxidative phosphorylation. Mitochondria generate both energy and ROS, even under normal conditions. Damaged mitochondria produce more ROS, contributing to cellular toxicity and potentially leading to organ damage. Dealing with and removing damaged mitochondria and maintaining mitochondrial quality are essential for ensuring cellular and organ homeostasis and protecting their structure and function. Mitochondrial function is regulated by dynamics such as fusion, fission, movement along microtubules, and interactions with organelles like the endoplasmic reticulum, lysosomes, lipid droplets, and melanosomes [1]. Proteins that regulate mitochondrial dynamics are integrated into cellular signaling pathways. For instance, the mitochondrial fission factor Drp1 is regulated by various post-translational modifications that interact with cellular processes like apoptosis, Ca^2+^ signaling, hypoxia response, and the cell cycle [2]. Recent evidence indicates that interactions between mitochondria and other organelles significantly contribute to mitochondrial quality control and functional regulation. Thus, the disruption of mitochondrial function can lead to various diseases [3].

Mitochondria have cell-type-specific phenotypes, perform dozens of interconnected functions, and undergo dynamic and often reversible physiological recalibrations [4]. In the brain, mitochondria are abundant in neuronal dendrites and synaptic terminals due to the high energy demands of neurons, which cannot store ATP [5]. Mitochondria can change their shape dynamically through controlled processes of fusion and fission. Mitochondria can also move actively between different parts of the cell, such as the cell body, axon, and presynaptic terminals [6]. Thus, adapting the energy supply to meet demand and maintain mitochondrial health is essential for cellular homeostasis and proper neuronal function [7], such as synaptic plasticity [8] and neurogenesis [9]. Mitochondria also play a central role in synthesizing biomolecules, including neurotransmitters [6] and hormones [10]. Based on these points, mitochondrial activity is crucial for modulating neuronal activity, short- and long-term plasticity, cellular resilience, and behavioral adaptations [11,12]. Mitochondria also interact with the endoplasmic reticulum (ER) at specific sites known as mitochondria-associated ER membranes, which cover 5–20% of the mitochondrial surface. Similar to how mitochondria create a network throughout neurons, the ER also forms a distributed network. Interactions at these contact sites are vital in regulating various physiological processes, including ATP production, autophagy, mitochondrial movement, and apoptosis [13]. Mitochondria are actively transported to and from synapses to supply energy and buffer Ca^2+^ levels, which is critical for sustaining synaptic activity. At presynaptic sites, they provide ATP to support prolonged synaptic function and buffer Ca^2+^ signals, thereby modulating neurotransmission and potentially limiting synaptic activity. The dynamic regulation of mitochondrial presence at synapses may also enhance computational flexibility [7]. Mitochondrial dysfunction leads to ROS synthesis and neuronal damage, which cause brain aging [14], neurodegenerative diseases [15], and psychiatric disorders [16], including bipolar disorder (BD) [17].

BD is a mental illness marked by extreme mood swings, alternating between depressive and manic episodes. Bipolar I disorder is defined by distinct manic episodes, which often include symptoms such as excessive confidence, grandiosity, increased talkativeness, severe disinhibition, irritability, a reduced need for sleep, and an intensely elevated mood. Psychotic features, such as delusions and hallucinations, are present in up to 75% of manic episodes and often disrupt psychosocial functioning to the point of requiring hospitalization. In contrast, bipolar II disorder is mainly characterized by recurrent depressive episodes alternating with hypomanic episodes instead of full mania [18]. BD has a global lifetime prevalence of 2.4% and a 12-month prevalence of 1.5% [19]. Pharmacological treatment, including mood stabilizers like lithium, valproate, and lamotrigine, along with antipsychotic agents, is the primary treatment approach. Clinicians adjust medications based on the patient’s manic, depressive, or remission phases. Some patients may not respond to these pharmacological treatments, potentially resulting in cognitive impairment and social dysfunction [20]. Despite numerous studies, predicting treatment response and prognosis in BD remains challenging [21,22,23]. BD ranks as the 17th leading source of disability among all diseases worldwide [24]. Approximately 6 to 7% of individuals with BD commit suicide, with studies showing that their suicide rates are 20 to 30 times higher than those in the general population [25]. Although BD has significant social impacts, its exact causes remain unknown. Mitochondrial dysfunction has been proposed as one possible explanation [17,26]. Evidence supporting this hypothesis includes creatine uptake studies using magnetic resonance spectroscopy in BD [27,28], postmortem brain studies [29,30,31], and the frequent comorbidity of BD with mitochondrial diseases [32]. Numerous reports have supported the involvement of mitochondrial dysfunction in BD’s pathophysiology [33,34], including its relationship with ROS [35,36]. An excess of ROS caused by an imbalance between oxidant production and antioxidant defenses is a key etiological factor in BD.

In this review, we will explore how different mitochondrial dysfunctions—such as energy and ROS metabolism, mitochondrial quality control regarding mitophagy, and calcium signaling regarding synaptic plasticity—lead to ROS production. We will also examine how these mitochondrial dysfunctions have been reported in BD and review possible links among BD, mitochondrial dysfunctions, and ROS. Additionally, we will discuss potential novel therapeutic targets to improve these conditions.

### Take-Home Message

•Mitochondria exhibit cell-type-specific traits, perform interconnected functions, and undergo dynamic, reversible physiological adjustments.•In neurons, mitochondria help maintain homeostasis by regulating energy production, ROS metabolism, calcium regulation, apoptosis, synaptic plasticity, and neurogenesis, which supports and protects neuronal function.•Multiple studies support the role of mitochondrial dysfunction in BD’s pathophysiology, especially its connection with ROS.

We mainly focus on energy and reactive oxygen species metabolism, mitophagy, synaptic plasticity, and lipid metabolism, discussing their relationships with ROS and BD. Table 1 provides an overview of these topics.

## 2. Energy and ROS Metabolism

Mitochondria are essential organelles responsible for energy production through aerobic respiration. Mitochondria consist of an outer and inner membrane, forming two distinct intracellular compartments: the mitochondrial matrix and the intermembrane space. The outer membrane is permeable to solutes smaller than 5 kDa, while the inner membrane is mainly impermeable. Enzymes of the citric acid cycle carry out a series of redox reactions within the mitochondrial matrix, generating reduced electron carriers such as NADH and FADH_2_. Complex I oxidizes NADH, and complex II oxidizes succinate, reducing ubiquinone to ubiquinol. Complex III then oxidizes ubiquinol, reducing cytochrome c. In complex IV, cytochrome c is oxidized, transferring electrons to molecular oxygen, which is reduced to water. During these processes, the energy from electron transport is utilized to pump protons from the matrix into the intermembrane space. These processes create a proton gradient across the inner mitochondrial membrane. The protons gradient drives ATP synthesis by complex V, which phosphorylates ADP to produce ATP [55].

ROS are typically generated as byproducts of the electron transport chain (ETC), mainly due to electron leakage at complexes I and III. During normal mitochondrial metabolism, only a small fraction of electrons escape through the ETC and reduce oxygen to form superoxide anions. These superoxide anions are then dismutated by SOD into hydrogen peroxide. Superoxide and hydrogen peroxide are key molecules of ROS with distinct chemical properties affecting their reactivity and diffusion. Superoxide, a negatively charged radical, forms stable salts with alkali metals but rapidly dismutates in aqueous solutions. Its reactivity is limited in water due to solvation and spontaneous disproportionation, but it can generate hydroxyl radicals via Fenton reactions [56]. Hydrogen peroxide, a neutral weak acid, decomposes into water and oxygen. Unlike superoxide, hydrogen peroxide can freely diffuse across lipid membranes, allowing for systemic distribution. In biological systems, superoxide’s reactivity is localized due to its limited diffusion, while hydrogen peroxide acts as a systemic oxidative signal or threat [57]. In the presence of reduced transition metals such as iron within the cell, hydrogen peroxide may further react to generate hydroxyl radicals. Hydroxyl radicals are highly reactive and can oxidize biomolecules such as proteins, lipids, carbohydrates, and nucleic acids. ROS are potent oxidizing agents, and cells possess antioxidant systems, including glutathione, SOD, and glutathione peroxidase, to neutralize them [58]. At low concentrations, mitochondrion-derived ROS, when properly regulated, act as signaling molecules and play a critical role in regulating essential cellular pathways. These pathways include metabolic adaptation, immune responses, and redox signaling [59].

Dysregulation of the ETC can lead to ROS production exceeding the capacity of the antioxidant network, exposing cells to oxidative stress. The brain, in particular, is susceptible to oxidative stress due to its high energy demands, presence of oxidizable polyunsaturated fatty acids, and relatively low antioxidant capacity. The impairment of mitochondrial function disrupts oxidative metabolism in neurons, potentially leading to symptoms such as psychosis and mood disturbances [60].

It is well established that oxidative stress significantly increases in patients with BD and is considered to mediate the neuropathological processes of BD [37,61]. Postmortem analyses of brain tissue have revealed dysfunction in glutathione, a tripeptide compound consisting of glutamic acid attached via its side chain to the N-terminus of cysteinylglycine, and elevated 4-hydroxynonenal, a major product of lipid peroxidation, in BD [61,62]. Antioxidant enzyme activity exhibits different behaviors depending on the phase of BD. For example, serum SOD activity has been reported to be higher during manic and depressive phases compared to levels in the remission phase and control groups [40]. The number of manic episodes is associated with elevated blood levels of 8-OHdG, an indicator of DNA oxidation, in BD [38]. Additionally, Yumru et al. found that serum SOD levels were significantly elevated in BD and antidepressant-induced mania patients compared to control groups [41]. These findings suggest that this may represent a compensatory response to prevent oxidative damage. The imbalance between oxidant production and antioxidant defenses is also characterized by catalase, protein carbonyl, glutathione peroxidase, 3-nitrotyrosine, lipid peroxidation, nitric oxide, and DNA/RNA damage, as evidenced by peripheral biomarkers and postmortem brain studies in BD [39,63,64]. It should be noted that this relationship is not necessarily caused by ROS originating from mitochondria, and other cellular sites of ROS production may certainly contribute or even be the main contributors. Also, there is evidence that ROS produced in mitochondria do not contribute to the oxidation of chromosomal DNA [65]. A study demonstrated an integrative approach using both in vitro (cell culture) and in vivo (mouse) experiments to develop a new model of mild mitochondrial dysfunction commonly reported in BD. The experiments showed consistent results between cell cultures exposed to rotenone, a mitochondrial complex I inhibitor that causes an extreme alteration in mitochondrial homeostasis and blocks autophagic flux by promoting an increase in ROS [66], causing significant cellular alterations and BD-like behavioral changes similar to those observed in BD patients [67]. Impaired autophagy and a very mild increase in ROS levels are related to a predisposition to manic-like behavior. Among the autophagy enhancers and ROS scavengers tested, lithium, a first-line drug for BD, is the most effective in counteracting rotenone-induced changes [68]. These findings suggest that mitochondrial dysfunction in BD is at least partially associated with mtROS.

Abnormalities in mitochondrial ETC complexes have also been observed in BD patients. Postmortem analyses of the brains of BD patients have shown reduced mRNA levels involved in the ETC. Specifically, eight genes coding for the components of the mitochondrial ETC, *NDUFS7* (NADH-ubiquinone oxidoreductase 20-kd subunit), *NDUFS8* (NADH-ubiquinone oxidoreductase 23-kd subunit), *UQCRC2* (ubiquinol-cytochrome C reductase complex core protein 2), *COX5A* (cytochrome c oxidase polypeptide Va), *COX6C* (cytochrome c oxidase polypeptide Vic), *ATP5C1* (ATP synthase gamma chain), *ATP5J* (ATP synthase coupling factor 6), and *ATP5G3* (ATP synthase lipid-binding protein), were identified. The downregulation of NADH-ubiquinone oxidoreductase 20 kd subunit (ETC complex I), cytochrome c oxidase polypeptide Vic (ETC complex IV), and ATP synthase lipid-binding protein (ETC complex V) were further verified by real-time PCR [43]. Furthermore, the dysfunction of the mitochondrial respiratory chain (MRC) complex I and decreased mRNA levels encoding complexes I through V have been identified in BD patients’ hippocampi and prefrontal cortices [42,43,69]. Notably, during manic episodes in BD patients, the expression of complex I-related genes (NDUFV1, NDUFV2, NDUFS1) in blood samples was significantly higher than that in control groups, suggesting a potential compensatory response [44]. Similarly, increased activity of ETC complex I was observed in the platelets of BD patients, whereas the activities of complexes II and IV were decreased [45].

These findings suggest that BD is a disorder characterized by mitochondrial dysfunction partially caused by mtDNA mutations and excessive ROS production [70]. MtDNA is particularly vulnerable to oxidative damage due to the lack of histone protection [71]. Also, the lack of non-coding regions in mtDNA increases the potential mutagenicity of oxidative damage [72]. ROS-induced damage is thought to create a vicious cycle that exacerbates mitochondrial dysfunction and increases oxidative stress [73]. Significant associations between DNA damage and BD, as well as between DNA damage and the severity of BD and depressive symptoms, have also been reported [46], indicating the need for further research.

## 3. Mitophagy

Oxidative stress involved in the pathobiology of mental disorders occurs when the generation of ROS exceeds the cell’s antioxidant capacity [74]. Under normal physiological conditions, ROS contribute to maintaining low-level homeostasis and regulating physiological activities. However, under pathological conditions, this balance is disrupted, leading to a significant increase in ROS levels. A substantial proportion of excessive ROS are released from dysfunctional mitochondria as endogenous mitochondrial ROS (mtROS). These mtROS cause damage to mitochondria, triggering a vicious cycle of structural and functional damage. mtROS are suggested to damage mtDNA, and oxidatively damaged mtDNA may trigger stronger inflammatory responses. Additionally, mtROS may promote the opening of mitochondrial permeability transition pores (mPTPs), facilitating the release of intrinsic mitochondrial components and potentially exacerbating mitochondrial damage [75].

Mitophagy is a form of selective autophagy that plays a crucial role in mitochondrial quality control and maintaining homeostasis. Mitophagy helps prevent stress stimuli induced by endogenous mitochondrial components, energy metabolism disorders, and apoptosis by systematically removing damaged or dysfunctional mitochondria from the cell. Through mitophagy, cells can maintain a balanced and stable environment free from the potentially harmful effects of dysfunctional mitochondria [75].

Current studies distinguish between two forms of mitophagy: ubiquitin-regulated mitophagy and receptor-regulated mitophagy (Figure 1). Ubiquitin-regulated mitophagy is promoted by the extensive ubiquitination of mitochondrial surface proteins. Ubiquitin chains on the outer mitochondrial membrane are recognized by isolation membranes, which selectively engulf and degrade mitochondria within autophagosomes. Specifically, when mitochondria are damaged, mitochondrial inner membrane depolarization occurs. This depolarization disrupts the import of proteins through the translocase of the inner membrane complex, which relies on a membrane potential. As a result, depolarized mitochondria are unable to import PINK1 into the inner membrane, preventing its cleavage by the PINK1/PGAM5-associated rhomboid-like protease. Consequently, PINK1 accumulates on the outer membrane [76], where it activates PARKIN. PARKIN recruited to the target mitochondria induces the ubiquitination of outer membrane proteins. Subsequently, several autophagy receptors—such as sequestosome 1 (p62/SQSTM1), optineurin, neighbor of BRCA1 gene 1, calcium-binding and coiled-coil domain 2, and Tax1-binding protein 1—bind to ubiquitin-labeled mitochondrial outer membrane proteins. These receptors recruit microtubule-associated protein light chain 3 (LC3), enabling connections between the isolation membrane and mitochondria through adaptor proteins, facilitating selective mitochondrial encapsulation by autophagosomes [75]. As an alternative pathway, receptor proteins directly bind LC3 without requiring ubiquitination, which induces receptor-regulated mitophagy. Under hypoxic conditions, hypoxia induced the dephosphorylation of FUNDC1 and enhanced its interaction with LC3 for selective mitophagy [77]. In addition, hypoxia JNK1/2 (c-Jun N-terminal kinase 1/2) phosphorylates BNIP3 at Ser 60/Thr 66, which hampers the proteasomal degradation of BNIP3 and drives mitophagy by facilitating the direct binding of BNIP3 to LC3 [78].

BD is reported in several studies to be closely associated with mitochondrial dysfunction. As described above, the process of mitophagy is indispensable for mitochondrial quality control and homeostasis. Scaini et al. reported that in BD patients, the expression levels of mitophagy-related proteins such as PARKIN, PINK1, p62/SQSTM1, and LC3 are reduced, whereas 18 kDa translocator protein (TSPO) pathway proteins (TSPO and voltage-dependent anion channel (VDAC)) are increased at both the mRNA and protein levels. TSPO is a protein located on the outer mitochondrial membrane that regulates metabolism and apoptosis. It forms complexes with VDAC that interact with various proteins and involve numerous functions and processes, including cholesterol transport, mitochondrial metabolism, apoptosis, cell proliferation, Ca^2+^ signaling, oxidative stress, and inflammation [47].

Peripheral blood mononuclear cells from BD patients show significantly increased mRNA and protein expression levels of TSPO and VDAC compared to those from healthy controls. Furthermore, the TSPO/VDAC ratio is higher in BD patients than in controls [47]. An increase in the TSPO/VDAC1 ratio leads to increased production of mtROS. This relationship appears to be direct, as increasing TSPO/VDAC1 ratio has been shown to lead to decreased mitochondrial ATP production and increased ROS levels [79]. Although the exact mechanism is not fully understood, a hypothesis suggests that TSPO, via VDAC1, inhibits mitophagy by limiting the PARK2-dependent ubiquitination of mitochondria for the concomitant accumulation of ROS [80]. A negative correlation between TSPO levels and mitophagy-related proteins has been observed, along with a negative correlation between VDAC and p62/SQSTM1 or LC3 protein levels [47]. TSPO inhibits the mitochondrial translocation of p62/SQSTM1, suppressing LC3 translocation and reducing mitophagy pathway activity. This mitophagy regulation depends on the VDAC to which TSPO binds [79]. When TSPO and VDAC levels increase and the expression of TSPO rises relative to VDAC, excessive production and accumulation of ROS are observed. This situation disrupts the downstream activity of the PINK1-PARK2 pathway, inhibiting PARK2-mediated protein ubiquitination and suppressing the recruitment of p62/SQSTM1. As a result, mitophagy is impaired, leading to the accumulation of dysfunctional mitochondria. Additionally, the increase in mitochondrial ROS induced by an elevated TSPO ratio is suggested to activate protein kinase Cε via the Raf-1-MEK1/2-ERK1/2 pathway, further promoting TSPO expression. Thus, TSPO is a critical factor in mitochondrial quality control through mitophagy, and the ratio of TSPO to VDAC expression is essential for maintaining cellular homeostasis [80].

TSPO-derived mitochondrial ROS may link the activation of TSPO and VDAC, potentially inducing the activation of the mitochondrial apoptosis pathway [79]. ROS can release cytochrome c from cardiolipin in the inner mitochondrial membrane. Furthermore, ROS activate VDAC, releasing cytochrome c into the cytoplasm through VDAC. The release of cytochrome c from mitochondria forms the initial stage of activating the mitochondrial apoptosis pathway [81]. In patients with BD, elevated gene expression levels of NLRP3-related proteins (NLRP3, ASC, and pro-caspase-1) have been observed, along with increased levels of caspase-1 activity and inflammatory cytokines IL-1β and IL-18. These findings suggest a strong positive correlation between the activation of the NLRP3 inflammasome and TSPO-associated proteins [47]. These observations imply that in BD patients, the elevation of the TSPO-VDAC complex not only leads to a reduction in mitophagy-related proteins but also contributes to the activation of apoptosis and increased inflammation through the activation of the NLRP3 inflammasome [47].

Additionally, in BD patients, excessive mitochondrial fission reduces mitophagy-related proteins and increases caspase-3 levels, leading to dysfunctional mitochondria, oxidative stress, and apoptosis. Studies investigating protein levels of apoptosis-promoting and -inhibiting factors in peripheral blood mononuclear cells have shown that BD patients exhibit significantly reduced levels of anti-apoptotic proteins, such as Bcl-xL, survivin, and Bcl-xL/Bak dimers. In contrast, pro-apoptotic protein active caspase-3 levels are significantly increased [47]. Moreover, apoptosis-related genes like FAS, BAK, and APAF-1 are upregulated in BD patients’ hippocampi [48]. Furthermore, peripheral blood mononuclear cells from BD patients display a reduced expression of the mitochondrial fusion-related proteins Mfn2 and Opa1, along with increased levels of the fission protein Fis1 [82].

These findings suggest that in BD patients, an imbalance in mitochondrial fission and fusion leans toward fission, accompanied by decreased mitophagy-related proteins and increased levels of caspase-3 protein. This excessive mitochondrial fission generates dysfunctional mitochondrial fragments, leading to altered mitochondrial morphology, increased oxidative stress, and apoptosis-mediated cell death [82]. As a result, apoptosis emerges as a predominant pathway for minimizing tissue damage in BD [48]. Postmortem studies show evidence of tissue-level cell death and atrophy in BD brains, including reduced neuron and glial cell density in the dorsolateral prefrontal cortex [83]. Additionally, cortical thinning, particularly in the frontal lobes, superior temporal, and temporoparietal regions, has been reported, along with bilateral reductions in subcortical volume, affecting the hippocampus, left thalamus, right nucleus accumbens, left cerebellar cortex, and brainstem. BD brains also exhibit ventricular enlargement [84]. Thus, the tissue-level alterations observed in BD brains likely result from a complex interplay of mitochondrial dysfunction, oxidative stress, and apoptosis.

## 4. Synaptic Plasticity and Abnormalities in Calcium Signaling

Mitochondria are critical for neuronal development and the regulation of synaptic plasticity [85]. The mechanisms underlying connectivity changes within the plasticity framework can be categorized into molecular and cellular levels. Molecular mechanisms, mediated by signaling pathways, drive gene transcription and protein synthesis essential for plasticity. These proteins induce cellular changes, which are classified into structural and functional plasticity. Structural plasticity encompasses neuronal changes such as neurogenesis and dendritic remodeling, while functional plasticity involves modifications in neuronal function, including long-term potentiation or depression, ultimately contributing to synapse formation [86].

We discuss synaptic plasticity from two perspectives: structural plasticity and functional plasticity. The structural plasticity of dendritic spines during synapse formation depends on the mitochondrial number within dendrites. Key processes like mitochondrial fission and fusion, regulated by Drp1 and OPA1, respectively, play significant roles. Overexpressing the dominant-negative form of Drp1 reduces mitochondrial density in dendrites, inhibiting synapse formation. In contrast, treating cells with creatine increases mitochondrial mass and activity, doubling synapse numbers. These findings underscore the indispensable role of mitochondria in structural synaptic plasticity [87,88,89,90]. From the perspective of functional plasticity, mitochondria are heavily involved in long-term potentiation (LTP), a crucial process for synaptic activity, learning, and memory [91,92]. Notably, ROS like superoxide and hydrogen peroxide are essential for LTP induction [93,94]. Mitochondria supply the necessary energy for synaptic signal transmission and plasticity by moving within synapses. Their numbers are regulated through dynamic fission and fusion processes [95,96]. However, increased oxidative stress can lead to abnormal mitochondrial fission, halted movement, and impaired functional plasticity [97].

Regarding synaptic formation, the functions of presynaptic axons and postsynaptic dendrites are crucial. The functions of both presynaptic axons and postsynaptic dendrites are significantly influenced by changes in intracellular Ca^2+^ levels resulting from metabolic and oxidative stress. Mitochondria act as Ca^2+^ sensors, sequestering Ca^2+^ after synaptic stimulation to support neurotransmission [98]. Thus, the disruption of calcium homeostasis due to mitochondrial dysfunction results in synaptic dysfunction and subsequently impairs synaptic plasticity.

One potential cause of abnormalities in Ca^2+^ homeostasis is mitochondrial membrane potential (MMP) dysfunction [49]. BD patients exhibit elevated intracellular Ca^2+^ levels across all mood states, with particularly high levels during manic episodes [99,100]. Additionally, changes in the expression of genes involved in Ca^2+^ signaling, neuroactive ligand–receptor interactions, and protein kinase PKA/PKC signaling pathways have been observed in BD patients [50]. Calcium/calmodulin-dependent protein kinase kinase 2 (CaMKK2) is a central component of the Ca^2+^–calmodulin (Ca^2+^-CaM)-dependent signaling pathway in neurons [101]. BD has been associated with mutations that affect the function or expression of CaMKK2 [51]. Furthermore, increased Bcl-2 enhances energy-dependent Ca^2+^ uptake capacity, improving resistance to mitochondrial dysfunction caused by excessive Ca^2+^ influx [102]. Thus, the levels of Bcl-2 family proteins play a crucial role in maintaining calcium homeostasis [103,104,105]. Notably, Bcl-2 is associated with the severity of manic symptoms in BD in a manic phase [106].

It is also known that mitochondria regulate synaptic plasticity by mediating part of the effects of glutamate and brain-derived neurotrophic factor (BDNF) [48]. BDNF is involved in all levels of neural plasticity, including neurogenesis and dendritic growth, and is considered a key regulator of synaptic plasticity [107]. Alterations in neurotrophic signaling, particularly involving BDNF, have been reported in BD. Decreased BDNF levels have been observed during manic and depressive episodes, along with transcriptional changes in the BDNF gene influenced by stress, neurofunctional impairments, and mood states [108,109,110]. A decrease in CaMKK2 function leads to reduced BDNF expression [48].

Glutamate is the most abundant excitatory neurotransmitter in the brain and is known to regulate synaptic plasticity through various mechanisms [111]. Changes in glutamatergic neurotransmission are implicated in the pathophysiology of BD. Numerous studies have reported elevated glutamate levels in BD patients’ plasma, serum, and cerebrospinal fluid [112]. Magnetic resonance spectroscopy and postmortem studies have also revealed increased glutamate levels in various brain regions of BD patients [113,114]. This relationship highlights the integrative role of mitochondria in linking neurochemical signaling pathways to synaptic function, with significant implications for understanding and treating BD.

In summary, mtROS play a role in synaptic plasticity and calcium signaling abnormalities. They act both as a contributing factor and as a consequence of other underlying mechanisms involved in the disorder. Further research is needed to clarify these complex interactions.

## 5. Lipid Metabolism

Metabolomics provides a real-time molecular snapshot of an organism’s physiological state, highlighting the interactions between genetic and environmental factors [115]. As a result, metabolomics reflects dynamic metabolic processes and biochemical changes, allowing for a more immediate understanding of disease mechanisms.

In the context of BD, metabolome analysis has revealed connections between mitochondrial dysfunction and the production of ROS. For example, lower levels of lactate and acetoacetate in the serum of BD patients suggest impaired mitochondrial energy production, potentially increasing ROS due to inefficient electron transport chain activity [116]. Systematic reviews showed that metabolomic studies consistently identify dysregulation in glucose, amino acid, and lipid metabolism, which are critical for mitochondrial function [117,118].

Recently, large-scale genome-wide association studies on BD have implicated the involvement of the fatty acid desaturase (FADS) locus [119,120,121]. Expression quantitative trait locus analysis suggested that the other major haplotype (haplotype A) conferring susceptibility to BD was associated with decreased expression of *FADS1/2* and likely lower enzyme activity [52]. FADS1 and FADS2 are involved in the metabolism of omega-3 fatty acids such as eicosapentaenoic acid (EPA) and docosahexaenoic acid (DHA). The role of FADS1 in fatty acid metabolism, including long-chain fatty acid metabolic processing, is essential for functional activity in mitochondria [53]. In addition, polyunsaturated fatty acids, such as EPA and DHA, can regulate mitochondrial membrane structure and function [122]. DHA also increased SOD2 activity in the cerebrum [123]. The suppression of FADS1 induces ROS generation through the mitochondrion-mediated apoptosis pathway [54] and ferroptosis pathway [124]. These findings suggest the interplay between lipid metabolism and ROS in BD.

## 6. Novel Therapeutics

### 6.1. Polyunsaturated Fatty Acids

Heterozygous *Fads1*/*2*-knockout mice showed hyperactivity and hypoactivity. The administration of the mood stabilizer lithium, supplementation with DHA, or a mixture of DHA and EPA prevented these episodic behavioral changes [125]. Meta-analysis and randomized control trials showed that EPA and DHA supplements potentially benefit BD [126,127,128]. Therefore, these results suggest that EPA and DHA could potentially be therapeutic candidates for BD. Further large-scale and high-quality clinical trials are needed to conclude that the intake of EPA and DHA effectively treats BD.

### 6.2. Autophagy Enhancers

Studies on the PINK1 pathway [129,130] suggest that this pathway selectively regulates the turnover of dysfunctional mitochondrial fragments. This mechanism may allow for the targeted removal of mitochondria with high levels of pathogenic mtDNA in heteroplasmic mtDNA diseases, potentially reducing heteroplasmy [131]. Aberrant cell function caused by oxidative stress and mitochondrial dysfunction may be ameliorated by enhancing autophagy [132]. Accumulated dysfunctional mitochondria may contribute to the pathophysiology of BD [26]. Thus, drugs that stimulate autophagy and mitophagy, the processes that degrade mitochondria, could serve as potential treatments for BD. In fact, lithium, carbamazepine, and valproate, already used in BD treatment, have been shown to enhance autophagy. Lithium enhances autophagy via two pathways: the inhibition of GSK3β and inositol monophosphatase. Lithium inhibits GSK3β, leading to elevated Bif-1 levels and interaction with the Beclin-1-VPS complex, which induces autophagy [133]. Lithium also inhibits inositol monophosphatase, depleting free inositol and reducing IP3 levels, thereby enhancing autophagy [134]. Carbamazepine activates AMP-activated protein kinase (AMPK) by decreasing inositol and IP3 levels. Once AMPK is activated, it inhibits mTORC1 phosphorylation, enabling ULK1 to interact with and be phosphorylated by AMPK. Activated ULK1 initiates autophagy [135]. Valproate activates AMPK and also inhibits histone deacetylase. Histone deacetylase inhibition increases histone acetylation, disrupting histone binding to nuclear DNA. The flow increases DNA susceptibility to transcription and may inhibit mTORC1 signaling, enhancing autophagy [136,137].

As a novel therapeutic candidate among substances that induce autophagy, Kara et al. showed that rapamycin, which is an mTOR inhibitor and autophagy enhancer, reduced mania-like aggression and reward-seeking behaviors in mice [138]. Damri et al. designed a novel mouse bipolar disorder-like model using the chronic administration of a low dose of the oxidative phosphorylation complex I inhibitor rotenone. Using the model mice, they found that the mitophagy enhancers lithium, trehalose, and resveratrol ameliorated the behavioral and neurochemical consequences of mild neuronal mitochondrial dysfunction [68].

### 6.3. Mitochondrial Modulators

Mitochondrial modulators are emerging as promising treatments for BD due to their role in enhancing mitochondrial function. These modulators reduce ROS production and mitigate oxidative stress, countering the disorder’s pathophysiology. Targeting mitochondria may stabilize cellular energy balance, reduce oxidative damage, and contribute to managing BD symptoms. In preclinical studies, therapeutics targeting mitochondrial function, such as coenzyme Q10, creatine, and compounds that enhance energy production, have been studied, and BD symptoms have been improved [139,140]. In addition, substances are currently being developed and undergoing clinical trials to treat mitochondrial diseases. These may also be candidate drugs to treat BD [141].

### 6.4. Nutraceuticals

Nutraceuticals such as N-acetylcysteine (NAC), which supports antioxidant function, and S-adenosylmethionine (SAMe), which influences neurotransmitter synthesis, have shown candidate molecules in managing BD. NAC, in particular, has been studied for its ability to reduce oxidative stress and stabilize mood [142]. A series of double-blind, randomized controlled trials investigated the effects of NAC as an adjunct treatment for BD, particularly targeting depressive symptoms [143,144]. In studies of patients in the maintenance phase (2 g/day), NAC lead to an improvement in depressive symptoms, manic symptoms, and quality of life without affecting cognitive function [145]. However, results varied, with some studies observing no significant difference from placebo, potentially due to high placebo response rates [146]. Additionally, combination treatments, such as NAC with aspirin or nutraceuticals, showed some enhanced response but did not significantly impact inflammatory markers (e.g., IL-6 and CRP) [147]. Findings suggest NAC may benefit specific subgroups but requires further investigation in larger trials to confirm its efficacy in BD treatment.

### 6.5. Epigenetic Modifiers

Epigenetic changes, such as histone acetylation and DNA methylation alterations, have been observed in BD. Histone deacetylase (HDAC) inhibitors, which generally result in increased histone acetylation, which is associated with a more open chromatin structure and increased transcriptional activity, are being explored as potential therapies [148]. HDAC inhibitors upregulate antioxidant enzymes and increase the expression of superoxide dismutase and catalase, thereby reducing ROS levels and enhancing defense against oxidative stress [149]. HDAC inhibitors play a role in reducing ROS production by improving mitochondrial function. They normalize oxidative phosphorylation within mitochondria, reduce electron leakage, and thus lower ROS production. Studies suggest that HDAC inhibitors help sustain ATP production and provide protection against oxidative stress [150].

Preclinical studies have investigated HDACs as potential therapeutic targets for BD, with HDAC inhibitors like hydroxamates, sodium butyrate, and valproate showing promise as candidate molecules. Valproate inhibits HDAC activity in vitro and may offer neuroprotection [149]. In animal models, HDAC inhibitors blocked histone deacetylation and exhibited antidepressant-like effects [151], suggesting a potential role in treating mood disorders. Studies indicate that HDAC5 downregulation may be crucial for the therapeutic efficacy of imipramine in depression models, implying that targeting HDAC5 could yield new antidepressant options [152]. However, specific HDAC inhibitors are in the early development stages, and challenges remain regarding potential side effects, blood–brain barrier penetration, and off-target effects [153]. Advances in epigenomic profiling could enable the precise targeting of HDAC subunits to refine BD therapies, as understanding these dynamic epigenetic modifications is critical for developing effective, individualized treatments.

### 6.6. Neurosteroid-Based Treatments

Steroid hormones significantly influence various central nervous system (CNS) functions. Circulating steroids from the adrenal glands, gonads, and placenta can cross the blood–brain barrier and impact CNS target cells. Additionally, the CNS itself is a highly steroidogenic environment that synthesizes steroids independently and metabolizes them from the bloodstream. Neurosteroid production is influenced by neuroinflammation, while steroids such as 17β-estradiol, dehydroepiandrosterone, and allopregnanolone can also modulate neuroinflammatory responses [154,155]. Neurosteroidogenesis starts with cholesterol transfer into mitochondria, where it is converted to pregnenolone and further metabolized into steroids like progesterone, testosterone, and estradiol [156]. Neurosteroids can regulate the inflammatory responses of microglia and astrocytes and have shown potential in regulating mood and cognition. Allopregnanolone, a neurosteroid, has recently been approved by the USA Food and Drug Administration for the treatment of post-partum depression [157] and is being explored for its potential role in BD due to its effects on GABAergic neurotransmission and mood regulation as well as the management of neuroinflammation [155,158].

### 6.7. Biologics and Immunotherapies as Anti-Inflammatory Agents

Growing evidence links neuroinflammation to BD. Novel therapies focus on reducing neuroinflammation, which may contribute to mood dysregulation. Biologics targeting specific immune pathways are being investigated for their potential to modulate the immune system’s impact on mood. Monoclonal antibodies targeting pro-inflammatory cytokines are being studied to determine if they can improve symptoms of BD. In 2013, Raison et al. conducted a double-blind randomized controlled trial with treatment-resistant depression, showing that infliximab, a biological TNF-α-inhibiting monoclonal antibody, could improve depressive symptoms with high baseline inflammatory biomarkers [159]. In 2019, McIntyre et al. evaluated infliximab in BD patients but did not use precise inflammatory subgrouping, which may explain the lack of confirmed efficacy. A post hoc analysis indicated that infliximab had a strong, lasting antidepressant effect, especially in those with a history of childhood physical abuse, highlighting the importance of precise patient subgrouping for treatment success [160]. Tocilizumab, a humanized anti-IL-6R monoclonal antibody used primarily for rheumatoid arthritis and cytokine release syndrome, may also be a therapeutic option for BD due to elevated IL-6 levels in BD patients. Clinical trials and case reports suggest that tocilizumab can alleviate symptoms like depression, mania, anxiety, and cognitive impairment, making it a promising personalized treatment for BD-related cognitive, mood, and mania symptoms [161].

## 7. Conclusions

We reviewed mitochondrial mechanisms and their link to dysfunction in BD and oxidative stress. We also discussed new treatments for BD targeting mitochondrial dysfunction and oxidative stress reduction. The pathophysiology of BD is complex, and it remains unclear which mitochondrial mechanisms are altered first or whether multiple mechanisms are affected simultaneously during disease progression. A deeper understanding of mitochondrial function and dysfunction in BD could clarify how current drugs work and guide the development of therapies targeting specific mitochondrial processes. Various studies have been conducted [162]. More are underway to determine whether pharmacological treatments for modulating mitochondrial function can improve the prognosis of BD and explore the potential benefits of antioxidant substances. Therapies enhancing mitochondrial function are expected to serve as primary or adjunctive treatments, shedding light on new therapeutic options for BD, especially for those who do not fully respond to standard therapies.

## Figures and Tables

**Figure 1 biomolecules-15-00383-f001:**
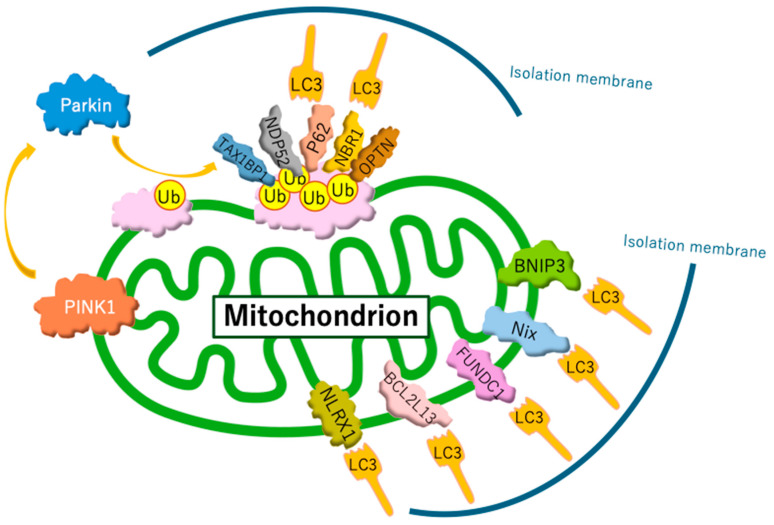
Mechanisms of mitophagy. Mitophagy involves two primary pathways: ubiquitin-regulated and receptor-regulated mechanisms. The PINK1/Parkin pathway is the most well-studied ubiquitin-regulated route. However, recent studies have identified additional ubiquitin ligases that perform functions similar to those of Parkin. In receptor-regulated pathways, certain receptor proteins can initiate mitophagy by directly binding to LC3, bypassing the need for ubiquitination.

**Table 1 biomolecules-15-00383-t001:** Mitochondrial dysfunction in bipolar disorder patients.

	Mitochondrial Dysfunction in Bipolar Disorder Patients
Energy and reactive oxygen species metabolism	Initial manic episodes are associated with increased oxidative stress and activated antioxidant defenses [37] Higher oxidative DNA damage correlates with the number of manic episodes [38] Increased levels of lipid peroxidation and DNA/RNA damage [39] Elevation of serum superoxide dismutase levels [40,41] Reduced expression of genes regulating oxidative phosphorylation and ATP-dependent proteasome degradation [42] Downregulation of mitochondrial electron transport chain genes linked to oxidative stress [43] Elevated mRNA levels of complex I-related genes during manic episodes [44] Decreased complex II and IV activities [45] Increased DNA and RNA damage by oxidation [46]
Mitophagy	Reduction in mitophagy-related proteins such as PARKIN, PINK1, p62/SQSTM1, and LC3 [47] The accumulation of damaged mitochondria surpasses the processing capacity of mitophagy [48]
Synaptic plasticity and abnormalities in calcium signaling	Dysfunction of mitochondrial membrane potential [49] Changes in the expression of genes involved in neuroactive ligand–receptor interactions and protein kinase PKA/PKC signaling pathways [50] Abnormalities in the function or expression of CaMKK2 [51]
Lipid metabolism	Decreased expression of FADS1/2 and likely lower enzyme activity [52] FADS1 in fatty acid metabolism is essential for functional activity in mitochondria [53] Suppression of FADS1 induces ROS generation through the mitochondrion-mediated apoptosis pathway [54]

## Data Availability

Not applicable.

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
