# Peer review of "The Association Among Bipolar Disorder, Mitochondrial Dysfunction, and Reactive Oxygen Species"

_biomolecules, 2025, doi:10.3390/biom15030383_

Round 1

Reviewer 1 Report (Previous Reviewer 2)

Comments and Suggestions for Authors

The author revise the review based on my suggestion, it looks organized now. 

Author Response

Comment: The author revise the review based on my suggestion, it looks organized now. 

Response: The authors would like to thank you for carefully reading our manuscript and for giving valuable suggestion. 

Reviewer 2 Report (New Reviewer)

Comments and Suggestions for Authors

This review addresses and debates the interesting topic of the relationship between bipolar disorder, mitochondrial dysfunction, and oxidative stress, discussing the specific mitochondrial dysfunctions observed in bipolar disorder and highlighting how these defects may contribute to the pathophysiology of the disorder. Potential therapeutic targets to improve mitochondrial function are also considered, which may pave the way for new treatments to manage or alleviate bipolar disorder symptoms.

I have no substantial criticisms to make on the review that I consider well written and in my opinion can be acceptable for publication after a minor revision.

As few minor points:

  • Section 5 of review on Lipid metabolism is too brief and concise, please expand it by describing the topic in a more extensive and explanatory way
  •  
  • Correct the mistake to line 174: the glutathione is not an enzyme but a tripeptide compound consisting of glutamic acid attached via its side chain to the N-terminus of cysteinylglycine

Author Response

This manuscript is a resubmission of an earlier submission. The following is a list of the peer review reports and author responses from that submission.

Round 1

Reviewer 1 Report

Comments and Suggestions for Authors

The review manuscript looks at associations between bipolar disorder and mitochondrial phenomena. Although the title indicates there will be a focus on mitochondrial ROS, this is not really the organizing principle around which the science is discussed. Further, the discussion of the science is often unsatisfying: there is too much text devoted to high school biochemistry; when new science is included, it is often just citing other people's reviews. Overall, the impression was that this review has little to add to the field. 

Major Comments

 1.     The title ‘The association between bipolar disorder and mitochondrial reactive oxygen species’ implies that the focus of this review is linking mtROS with pathogenesis of BD. While the authors do discuss how different types of mitochondrial dysfunction lead to increased ROS production, mtROS is rarely actually the focus. Rather, ROS production/oxidative stress are presented as symptoms of mitochondrial dysfunction alongside BD. While it is true that excess ROS/oxidative stress may cause cellular damage and diseases like BD, there are few mechanistic links provided between mitochondrial ROS and BD. Given the content of the review, I think it would be more appropriate for the title and introduction to frame this as an association between mitochondrial dysfunction and BD.

2.     Similarly, while the review understandably focuses on mitochondria and thus mitochondrial ROS, some results being presented don’t conclusively identify mitochondria as the source of ROS. For example, the authors describe studies where BD patients exhibited higher evidence of oxidative stress at the level of DNA damage, lipid peroxidation, serum SOD levels, etc. These symptoms are not necessarily caused by ROS originating from mitochondria, and other cellular sites of ROS production (of which there are many) may certainly contribute or even be the main contributors. In this vein, there is evidence that ROS produced in mitochondria does not contribute to oxidation of chromosomal DNA (van Soest et al., 2024). I think the manuscript would benefit from a more mechanistic link being made between ROS and the cellular/neurological pathophysiology of BD.

 3.     The sections describing therapeutic approaches targeting mitochondrial dysfunction in BD are interesting, but do not align with the scope implied by the title and introduction.

Minor Comments

 1.     Lines 146-151. It might be beneficial here to discuss the chemical properties of superoxide and hydrogen peroxide here. Given that SOD is presented here as an antioxidant/neutralizing enzyme, it should be clarified how the diffusion/reactive properties of each differ.

2.     Line 174. Comment on which genes; also mRNA or protein level? Mitochondrially encoded or genomic?

3.     Line 185. Also mention that the lack of non-coding regions in mtDNA increase potential mutagenicity of oxidative damage there.

4.     Line 215: Specify that it is transport of PINK1 through membrane import systems that is disrupted. As written, explanation is a bit vague.

5.     Line 228. Specify which stimuli/stressors. *Question for Dr. Stuart: is it ok to call them out for citing a review for this, where the review is citing a different review with more information?

6.     Line 254. Explain why increase in the TSPO/VDAC ratio causes mtROS production. Is this a direct mechanism, or is the ROS increase downstream.

7.     Lines 277-278. Not sure what the connection is between these findings and fusion/fission.

8.     Line 292. Again bad citation. Cites a review where they cite 2 primary articles for this information. Also I think this claim about apoptosis being the primary method for BD patients to clear dysfunctional mitochondria should be supported with data showing tissue-level cell death and atrophy in BD brains.

9.     Line 293. The section on synaptic plasticity is interesting and demonstrates the role of mitochondrial dynamics in synapse formation etc. However, there is very little implication of mtROS, other than how ROS are needed for long-term potentiation. It was also mentioned that oxidative stress can lead to impaired plasticity by interfering with mitochondrial fission and movement. As a result, it is not clear whether mtROS are contributing to BD, or are downstream of something else. Chicken and egg situation.

10.  Line 368. Paper cited here is in melanocytes, not brain. Would be good to comment more here on the roles of fatty acids in the brain specifically.

11.  Line 371. Interesting section. But little to nothing to do with mtROS, or even mitochondria.

12.  Line 394. Similar complaint, little to do with mtROS.

13.  Line 445: should specify that it’s countering the disease’s pathophysiology, not contributing to it.

14.  Line 469. Specify why HDAC inhibitors would increase gene transcription instead of just saying ‘modulate gene transcription’.

Reference

van Soest, D.M.K., Polderman, P.E., den Toom, W.T.F. et al. Mitochondrial H2O2 release does not directly cause damage to chromosomal DNA. Nat Commun 15, 2725 (2024). https://doi.org/10.1038/s41467-024-47008-x

Reviewer 2 Report

Comments and Suggestions for Authors

In this review, the Author review three topics: First,  how various types of mitochondrial dysfunction contribute to the production of ROS. Second, how abnormalities in calcium signaling, can lead to mitochondrial dysfunction and specific mitochondrial dysfunctions observed in BD. Third, potential therapeutic targets to improve mitochondrial function, which could pave the way for new treatments to manage or mitigate symptoms of BD.

First, Please label each section or topics make it easy to read. for example,1. backgorund, 2, Energy and ROS metabolism....

Second, In abstract, the author said"abnormalities in calcium signaling, which are crucial for synaptic plasticity, can lead to mitochondrial dysfunction.", I don't find a separate section for abnormalities in calcium signaling but with Synaptic plasticity. Please make it more clear in abstract.